# Extraneural infection route restricts prion conformational variability and attenuates the impact of quaternary structure on infectivity

Sheng Chun Chang[1], Maria Immaculata Arifin[1], Waqas Tahir[2], Keegan John McDonald[1], Doris Zeng[1], Hermann M. Schatzl[1,3,4], Samia Hannaoui[1], Sabine Gilch [1,3,4]*

1 Faculty of Veterinary Medicine, University of Calgary, Calgary, Canada, 2 Canadian and WOAH Reference Laboratory for BSE, Canadian Food Inspection Agency, Lethbridge, Canada, 3 Hotchkiss Brain Institute, University of Calgary, Calgary, Canada, 4 Snyder Institute for Chronic Diseases, University of Calgary, Calgary, Canada

* sgilch@ucalgary.ca

## Abstract

Prions can exist as different strains that consist of conformational variants of the misfolded, pathogenic prion protein isoform PrPSc. Defined by stably transmissible biological and biochemical properties, strains have been identified in a spectrum of prion diseases, including chronic wasting disease (CWD) of wild and farmed cervids. CWD is highly contagious and spreads via direct and indirect transmission involving extraneural sites of infection, peripheral replication and neuroinvasion of prions. Here, we investigated the impact of infection route on CWD prion conformational selection and propagation. We used gene-targeted mouse models expressing deer PrP for intracerebral or intraperitoneal inoculation with fractionated or unfractionated brain homogenates from white-tailed deer, harboring CWD strains Wisc-1 or 116AG. Upon intracerebral inoculation, Wisc-1 and 116AG-inoculated mice differed in conformational stability of PrPSc. In brains of mice infected intraperitoneally with either inoculum, PrPSc propagated with identical conformational stability and fewer PrPSc deposits in most brain regions than intracerebrally inoculated animals. For either inoculum, PrPSc conformational stability in brain and spinal cord was similar upon intracerebral infection but significantly higher in spinal cords of intraperitoneally infected animals. Inoculation with fractionated brain homogenates resulted in lower variance of survival times upon intraperitoneal compared to intracerebral infection. In summary, we demonstrate that extraneural infection mitigates the impact of PrPSc quaternary structure on infection and reduces conformational variability of PrPSc propagated in the brain. These findings provide new insights into the evolution of stable CWD strains in natural, extraneural transmissions.

## Author summary

Chronic wasting disease (CWD) is a prion disease spreading among wild and farmed cervids. Several strains of CWD prions have been identified that propagate stably upon intracerebral passage in experimental models. Understanding the emergence and adaptation of

**Data Availability Statement:** All relevant data are within the manuscript and its Supporting Information files.

**Funding:** This study was funded by the Alberta Prion Research Institute (grant #201800003 to SG and grant #201600023 to HMS) and the Natural Sciences and Engineering Council of Canada (NSERC; RGPIN-2019-05309 to SG). SG was supported through the Canada Research Chairs program. The funders had no role in study design, data collection and analysis, decision to publish, or preparation of the manuscript.

**Competing interests:** The authors have declared that no competing interests exist.

prion strains is critical for risk assessment of CWD cross-species transmission. We capitalize on our new gene-targeted mouse models of CWD that recapitulate key features of CWD pathogenesis in cervids. We infected these mice with whole or fractionated brain homogenates of deer containing previously characterized, distinct CWD strains by inoculation either into the brain or intraperitoneally. We found that the biochemical properties and pathological hallmarks of prions generated in central nervous system tissues of those mice differed, depending on route of inoculation. Furthermore, mouse survival times upon inoculation with fractionated brain homogenates were less variable upon intraperitoneal compared to intracerebral inoculation. Our findings shed new light on the impact of extraneural infection routes that mirror natural transmission and unravel peripheral infection as a selective barrier that restricts the conformational diversity of prions in the brain.

## Introduction

Prions, infectious agents solely comprised of protein, cause fatal neurodegenerative diseases such as Creutzfeldt-Jakob disease in humans, and various prion diseases in other mammals, including chronic wasting disease (CWD) in farmed and wild-living cervids [1–5]. With infected animals shedding significant amounts of infectious prions in excreta, bodily fluids, and tissues, CWD is highly contagious and is the most widespread and ecologically relevant prion disease [6–8]. Infection occurs through peripheral entry sites, followed by lymphoreticular replication and transport of prions to the central nervous system (CNS). CWD is found in 32 US states, 5 Canadian provinces, Norway, Finland, and Sweden [9]. The unknown zoonotic potential raises significant public health concerns. While there is no documented case of CWD in humans, and several studies indicate a strong transmission barrier [10–13], more recent evidence suggests that this barrier might not be absolute [14–17].

Prions replicate by conformational conversion of the host's cellular prion protein (PrP$^C$) into the disease-associated isoform PrP$^{Sc}$ [3]. Prion strains manifest as distinct disease phenotypes, characterized by distinct clinical presentations, distribution of PrP$^{Sc}$ and spongiform lesions in the brain, and biochemical properties of PrP$^{Sc}$. It has been postulated that prion strains are encoded by PrP$^{Sc}$ conformers [18–21]. Direct evidence for this concept has been provided recently by cryo-EM analysis, demonstrating strain-specific differences in the structure of PrP$^{Sc}$ [22–24].

Typically, prion strain characterisation involves intracerebral inoculation of prion-infected tissue homogenate and serial passaging in a model with the same genetic background, followed by biochemical analysis of PrP$^{Sc}$, and neuropathological assessment. Syrian hamsters, bank voles and transgenic mouse models overexpressing deer or elk PrP$^C$ were instrumental in identifying strains from natural CWD cases in North America and Europe [25–32]. These include Wisc-1, a strain isolated from an experimentally inoculated white-tailed deer, and 116AG, which contains a mixture of substrains different from Wisc-1 and originates from a wild-living white-tailed deer heterozygous for a PrP gene polymorphism at codon 116, resulting in an alanine to glycine substitution (A116G). Noteworthy, intracerebral inoculation circumvents the initial process of neuroinvasion, which describes how prions traverse through the host from the peripheral site of infection to the CNS. Interactions with the host's immune system and tissue-specific environments that can degrade or propagate prions [33–37], as well as by prion strain-specific properties like PrP$^{Sc}$ particle size or conformational stability [38–40] affect the efficiency of this process. These findings indicate that interactions during

neuroinvasion, which are absent in intracerebral infection, impose selective pressure on prions. We hypothesized that the route of infection affects PrP$^{Sc}$ conformational selection, and that efficiency of neuroinvasion depends on PrP$^{Sc}$ particle size.

Our group recently developed gene-targeted mouse models of CWD by replacing murine with deer wild-type PrP coding sequence. These mice are susceptible to peripheral inoculation and recapitulate CWD pathogenesis as observed in the cervid host, different from random integration transgenic mice [5]. Here, we used this model for intracerebral (i.c.) and intraperitoneal (i.p.) inoculation with Wisc-1 and 116AG CWD isolates from white-tailed deer or fractionated PrP$^{Sc}$ aggregates [26–29], followed by characterisation of PrP$^{Sc}$ properties and neuropathology. We demonstrate that i.p. inoculation abolishes differences in PrP$^{Sc}$ conformational stability and neuropathological hallmarks in the brain that are observed upon i.c. inoculation with Wisc-1 or 116AG prions. Our results indicate that route of inoculation and direction of prion transport in the CNS, i.e., retrograde versus anterograde, are key determinants of the similarity of PrP$^{Sc}$ conformational stability in brain and spinal cord tissues. Unexpectedly, low variance in survival times upon i.p. compared to i.c. inoculation of fractionated PrP$^{Sc}$ suggests that efficient pathogenesis upon peripheral infection might be independent of PrP$^{Sc}$ quaternary structure. Overall, our findings indicate that conformational and pathological differences between strains are abolished upon peripheral infection. This suggests that the process of neuroinvasion imposes selective pressure that reduces the diversity of prion conformations capable of stable propagation in the brain, while conformational diversity is still present in other host tissues including the spinal cord that are relevant in inter- and cross-species transmission.

## Results

### Survival times of *Prnp*.Cer.Wt mice inoculated intracerebrally and intraperitoneally with Wisc-1 and 116AG prions

To study the impact of route of inoculation on PrP$^{Sc}$ conformational selection, we inoculated *Prnp*.Cer.Wt mice with Wisc-1 and 116AG prions via i.c. and i.p. routes. Animals inoculated i.c. with Wisc-1 prions developed terminal prion disease at 457 ± 16 dpi, while those inoculated i.c. with 116AG prions reached the endpoint at 449 ± 9 dpi (**Fig 1A**; [5]). Mice inoculated i.p. reached the endpoint at 545 ± 14 dpi for Wisc-1 prions and 547 ± 21 dpi for the 116AG prions (**Fig 1A**; [5]). Survival times were significantly shorter upon i.c. inoculation compared to the i.p. route but remained congruent among the same route of inoculation regardless of inoculum. Animals inoculated i.c. with Wisc-1 prions accumulated significantly higher amounts of PK-resistant PrP$^{Sc}$ (PrP$^{res}$) than those infected i.c. with 116AG prions, despite similar survival times; however, it is important to note that PrP$^{res}$ does not reflect total PrP$^{Sc}$, and that PK-sensitive PrP$^{Sc}$ also might contribute to pathogenesis. In addition, i.p. inoculated mice had less PrP$^{res}$ in the brain than i.c. inoculated animals. This difference was more pronounced and statistically significant in Wisc-1 prion-inoculated mice (**Fig 1B and 1C**). To rule out that these quantitative differences in PrP$^{res}$ amount were due to overall PK resistance, we performed PK digestion kinetics and determined the concentration of PK needed to degrade 50% of PrP$^{Sc}$ (cPK$_{50}$). PrP$^{Sc}$ in brain homogenates of mice inoculated with the 116AG isolate had a higher cPK$_{50}$ compared to Wisc-1 inoculated mice, and the cPK$_{50}$ was similar for PrP$^{Sc}$ generated upon i.c. and i.p. inoculation (**S1 Fig**).

In summary, Wisc-1 and 116AG-inoculated gene-targeted mice had similar survival times depending on route of inoculation. Overall, survival times were not directly correlated to the amount of PrP$^{res}$ generated in their brains.

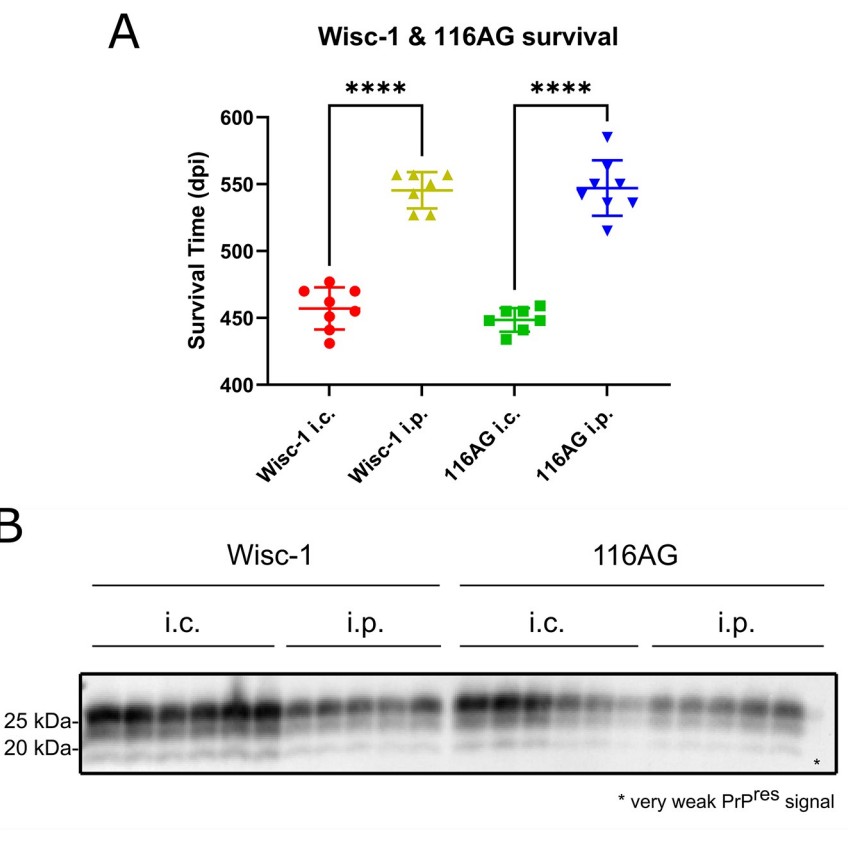

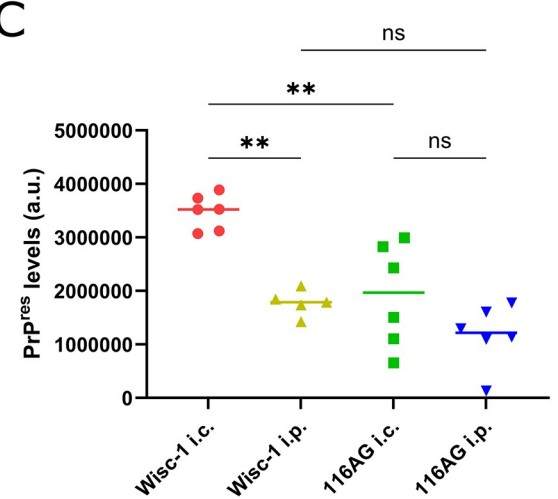

**Fig 1. *Prnp*.Cer.WT mice inoculated i.c. and i.p. with Wisc-1 and 116AG prions.** Statistical analysis was performed using one-way ANOVA followed by a Tukey's multiple comparison test. (A) Survival times of *Prnp*.Cer.WT mice challenged i.c. and i.p. with Wisc-1 and 116AG prions. The y-axis represents the survival period (dpi) and the x-axis the different inocula and routes of inoculation. **** $p$-value < 0.0001. Error bars: SD. (B) Western blot depicting PrP$^{res}$ signals (50 μg/ml PK) from Wisc-1 and 116AG prions inoculated individual animals through the two routes of inoculations. (C) Quantitative comparison of PrP$^{res}$ from i.c. vs. i.p. inoculated animals. Each point on the graph represents one mouse. Statistical analysis was performed using one-way ANOVA followed by a Tukey's multiple comparison test. Wisc-1 i.c. vs i.p. ** $p$-value = 0.0011, Wisc-1 i.c. vs 116AG i.c. ** $p$-value = 0.0016.

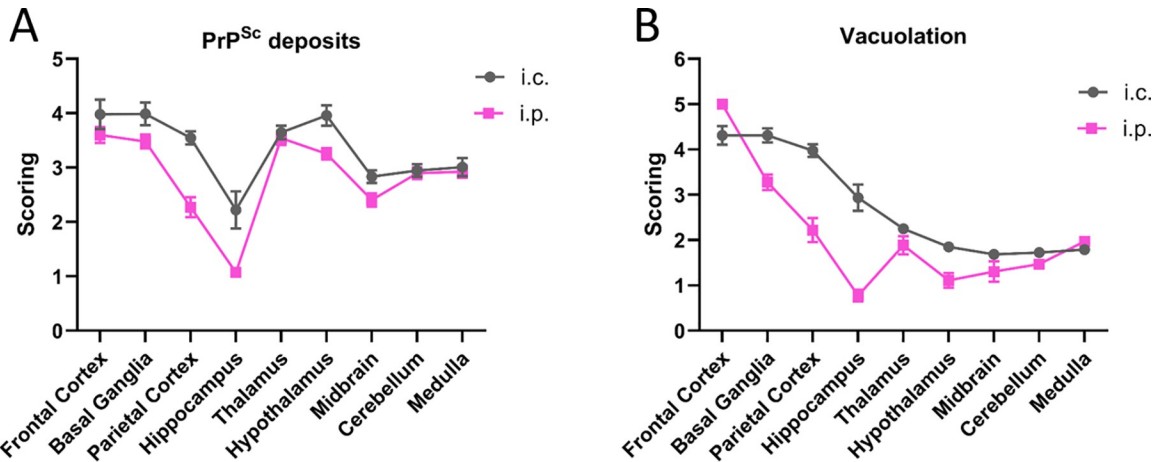

**Fig 2. Immunohistochemical and vacuolation analyses of *Prnp*.Cer.Wt mouse brain tissue inoculated i.c. or i.p.** (A) PrPSc deposition scoring comparison of mice inoculated i.p. or i.c. (B) Vacuolation scores comparison of mice inoculated i.p. or i.c. Brains of n ≥ 3 mice/route of inoculation were analysed. Error bars: SEM.

## PrPSc deposition and vacuolation profiles are different depending on inoculation route

Next, we analyzed brain sections to determine the PrPSc distribution and vacuolation profiles. Initial scoring of PrPSc deposits and vacuolation across nine brain regions indicated very similar profiles when comparing Wisc-1 and 116AG inoculated groups (S2 Fig). More variation between the two inocula was observed in i.c. than i.p. infected mice for both PrPSc deposits and vacuolation; however, the small number of mice analysed from each individual group does not allow us to draw conclusions about strain-specific differences. Therefore, we focused our analysis on comparing PrPSc deposits and vacuolation profiles between the two inoculation routes irrespective of inoculum (Fig 2). In brain sections of i.p. inoculated mice we observed markedly less PrPSc deposits in the parietal cortex and hippocampus than in i.c. inoculated animals (Figs 2A and S2). Minor differences in the extent of deposits were observed elsewhere in the brain except for the thalamus, medulla, and cerebellum, where there were no differences between the two routes of inoculations. Representative images of the frontal cortex, hippocampus, thalamus, and cerebellum are shown in S3 Fig.

Similar results were obtained for the vacuolation profiles. Brain sections of i.p. inoculated mice had less severe vacuolation in several brain regions compared to i.c. inoculated mice, with the most pronounced differences in basal ganglia, parietal cortex and hippocampus. (Figs 2B and S2). In the frontal cortex, thalamus and midbrain, vacuolation was very similar, and almost identical in the cerebellum and medulla between routes of inoculation (Figs 2B and S2).

Overall, both PrPSc deposition and vacuolation were more pronounced in specific brain regions of i.c. inoculated animals, irrespective of the CWD isolate used for inoculation.

## PrPSc conformational stability in brain and spinal cord is differently affected by route of inoculation

In order to determine PrPSc conformer diversity, we analysed PrPSc conformational stability in brain and spinal cord homogenates of the inoculated mice. We subjected tissue homogenates to denaturation with guanidine hydrochloride (GdnHCl) to determine the concentration needed to denature 50% of PrPSc (GdnHCl$_{50}$; representative dot blot shown in S4 Fig).

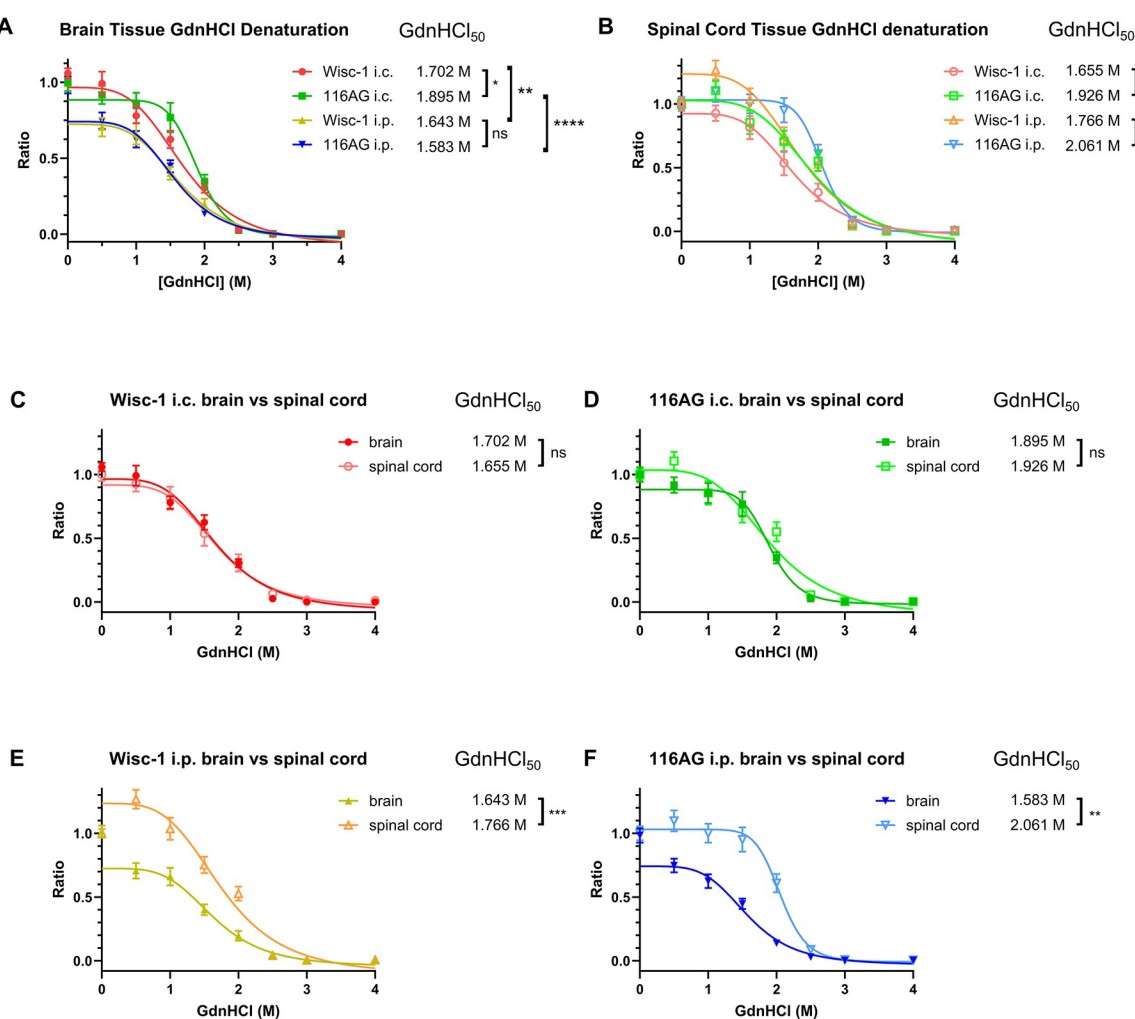

**Fig 3. Conformational stability of *Prnp*.Cer.Wt brain and spinal cord homogenates of mice inoculated i.c. or i.p. with Wisc-1 or 116AG prions.** Curve fitting, calculations and statistical analysis were performed through the GraphPad Prism 10 four-parameter sigmoidal equation and unpaired *t*-test. (A) Quantified brain homogenate PrP^res signals after guanidine denaturation, assessed by densitometric analysis as a ratio of the baseline signal at 0M GdnHCl. Wisc-1 i.c. vs 116AG i.c. * *p*-value = 0.0166. Wisc-1 i.c. vs Wisc-1 i.p. ** *p*-value = 0.0090. 116AG i.c. vs i.p. **** *p*-value < 0.0001. Wisc-1 i.p. vs 116AG i.p. ns *p*-value: not significant. (B) Quantified spinal cord homogenate PrP^res signals after guanidine denaturation, assessed by densitometric analysis as a ratio of the baseline signal at 0M GdnHCl. Wisc-1 i.c. vs. 116AG i.c. * *p*-value = 0.0185. Wisc-1 i.p. vs. 116AG i.p. * *p*-value = 0.0416. (C) Comparison of Wisc-1 i.c. brain vs. spinal cord homogenate PrP^res signals. ns *p*-value: not significant. (D) Comparison of 116AG i.c. brain vs. spinal cord homogenate PrP^res signals. ns *p*-value: not significant. (E) Comparison of Wisc-1 i.p. brain vs. spinal cord homogenate PrP^res signals. *** *p*-value = 0.0006. (F) Comparison of 116AG i.p. brain vs. spinal cord homogenate PrP^res signals. ** *p*-value = 0.0046. Error bars: SEM.

Intracerebral inoculation resulted in GdnHCl$_{50}$ values that were significantly different between Wisc-1 and 116AG prion-infected mice (**Fig 3A**). In contrast, conformational stability of PrP^Sc upon i.p. inoculation was nearly identical for Wisc-1 and 116AG prion-infected brain homogenates. Generally, a higher conformational stability of PrP^Sc was found for i.c. compared to i.p. inoculation with both isolates. Next, to investigate whether the conformational stability of prions is consistent across different regions of the CNS, we assessed the GdnHCl$_{50}$ values in spinal cord tissues. While conformational stabilities were not significantly different between route of inoculation for each inoculum, significant differences were observed between spinal cord tissue of i.c. and i.p. inoculated animals for both inocula (**Fig 3B**). Strikingly, i.c. inoculation resulted in closely aligned denaturation profiles in brain and spinal cord homogenates for

each inoculum (**Fig 3C and 3D**); however, in the spinal cords of i.p. inoculated animals, the PrP$^{Sc}$ conformational stability was significantly higher than in the brain, regardless of the isolate used for infection (**Fig 3E and 3F**).

## Survival times of mice infected i.p. are independent of PrP$^{Sc}$ aggregate complexity

Given the complex interactions and barriers that prions need to overcome during neuroinvasion, we questioned if the route of inoculation impacts the specific infectivity previously attributed to PrP$^{Sc}$ particle size or quaternary structure [41], as characterized by i.c. infection. To directly compare survival time profiles upon i.c. and i.p. inoculation of PrP$^{Sc}$ aggregates separated according to molecular weight, we subjected mouse brain homogenates infected with Wisc-1 or 116AG prions to sedimentation velocity ultracentrifugation [42–44]. We have used brain homogenates of 4$^{th}$ passage Wisc-1 and 116AG prions in tg(CerPrP132M)1536$^{+/+}$ mice [28,29,45] to minimize the risk of conformational or substrain mixtures in the inoculum. This will allow us to draw conclusions based on PrP$^{Sc}$ aggregate size or quaternary structure, with minimal, if at all, variation in tertiary structure. Gradient fractions with or without PK digestion were analysed by Western blot to determine PrP distribution profiles. For PK-resistant PrP$^{Sc}$, the profiles were slightly different, with fraction 10 for Wisc-1 and fraction 12 for 116AG harboring the highest amounts (**Fig 4A**). Even numbered fractions were inoculated i.c. or i.p. into *Prnp*.Cer.Wt mice. Survival times were plotted against fraction number and PrP distribution. In mice inoculated i.c. with either strain, we observed the shortest survival times in animals challenged with fraction 16 (**Fig 4A**). When comparing the survival times of i.c. to i.p. inoculated animals, it appeared that i.p. infected mice had overall more similar survival times among all fractions. To verify this, we compared the variance of survival times between the infection routes for each inoculum. For both strains, i.p. inoculated animals across all fractions had a significantly lower variance and smaller range, compared to i.c. inoculated mice (**Fig 4B**). Furthermore, we compared the distribution of PrP$^{Sc}$ deposits by IHC in the brains of animals inoculated i.p. or i.c. with representative fractions from the top (fractions 6 and 8), middle (fractions 16 and 18) or bottom (fractions 26 and 28) of the gradient. For Wisc-1 inoculated mice the distribution profiles were comparable within and between fractions, with i.p. inoculated mice generally harboring lower levels of PrP$^{Sc}$ deposits with the exception of cerebellum (**Fig 5A**). In brain sections of animals infected with fractionated 116AG prions, consistent with the results for Wisc-1, levels of PrP$^{Sc}$ deposits were lower in i.p. inoculated animals, except for cerebellum and medulla of top and middle fraction inoculated mice (**Fig 5B**). Distribution profiles closely aligned upon i.c. and i.p. inoculation of middle fractions. For top and bottom fraction groups, disparities were observed particularly in the hippocampus, and between i.p. and i.c. routes. Western blot analysis confirmed lower levels of PK-resistant PrP$^{Sc}$ in mice challenged i.p. with fractionated samples of either inoculum (**S5 Fig**).

In summary, our findings suggest that PrP$^{Sc}$ particle size has limited impact on the survival times of i.p. inoculated mice. Comparable to the outcomes observed in mice inoculated with unfractionated brain homogenate, brain sections of i.p. inoculated mice displayed reduced PrP$^{Sc}$ deposition across most brain regions, except for the cerebellum.

## Discussion

Several strains of CWD have been identified upon intracerebral passage of isolates from cervids in transgenic mice with varying expression levels of deer or elk PrP [25,26,28,29,31,45,46], or bank voles [32]. Such studies are instrumental to monitor the variety of PrP$^{Sc}$ conformations present in the host's brain; however, information about PrP$^{Sc}$ conformational selection

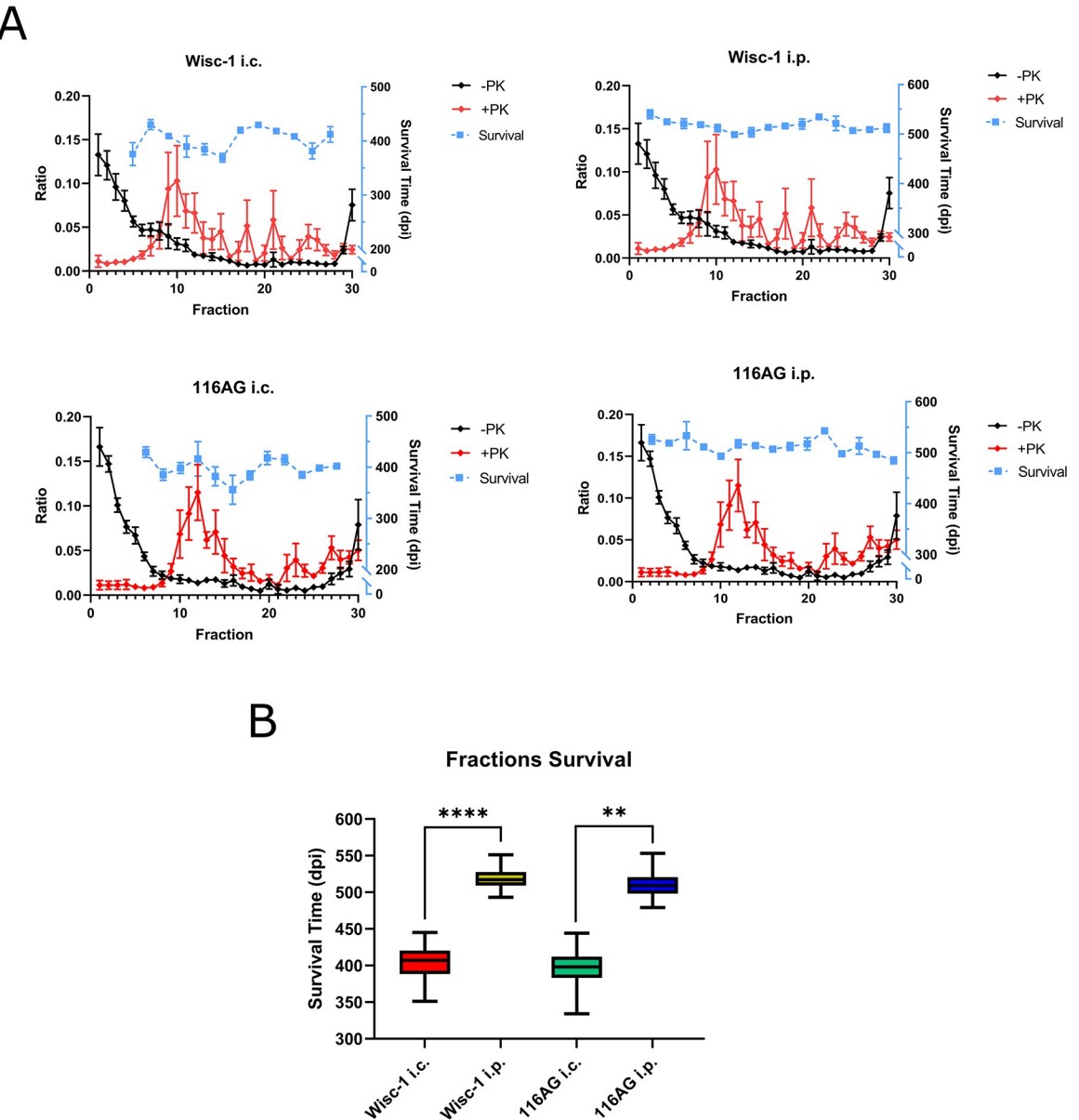

**Fig 4. Sedimentation profiles of CWD prion strains and survival times of the *Prnp*.Cer.Wt mice inoculated with prion fractions.**
(A) Wisc-1 and 116AG prions were solubilized and subjected to sedimentation velocity ultracentrifugation. Thirty Fractions were collected and quantified by Western blot with total PrP (-PK, black line) and PK-resistant (+PK, red line) depicted. Signals of every fraction were calculated as a ratio of the sum of signals from each respective replicate $n \geq 3$, error bars: SEM. Animals were subsequently inoculated i.c. or i.p. with alternating prion fractions from either isolate and survival times depicted with blue lines. Error bars for survival: SD. (B) Survival figures were pooled according to the different CWD prion strains and route of inoculation, and the box plot depicts the interquartile range (box) and range (whisker) of each group. F-test shows that the variance of the i.p. inoculated animals are significantly reduced than the i.c. inoculated animals (Wisc-1 i.c. vs. i.p. **** *p*-value < 0.0001, 116AG i.c. vs. i.p. ** *p*-value = 0.0094).

upon natural infection is limited, particularly for CWD prions, as the commonly used transgenic mouse models have very limited susceptibility to peripheral infection. Gene-targeted mouse models of CWD recently established by our group and others [5,47] preserve the authentic temporal and spatial expression of cervid PrP. These models accurately replicate CWD pathogenesis observed in cervids, including susceptibility to natural infection routes and prion shedding [5,47]. This now facilitates detailed investigations of PrP^Sc conformational

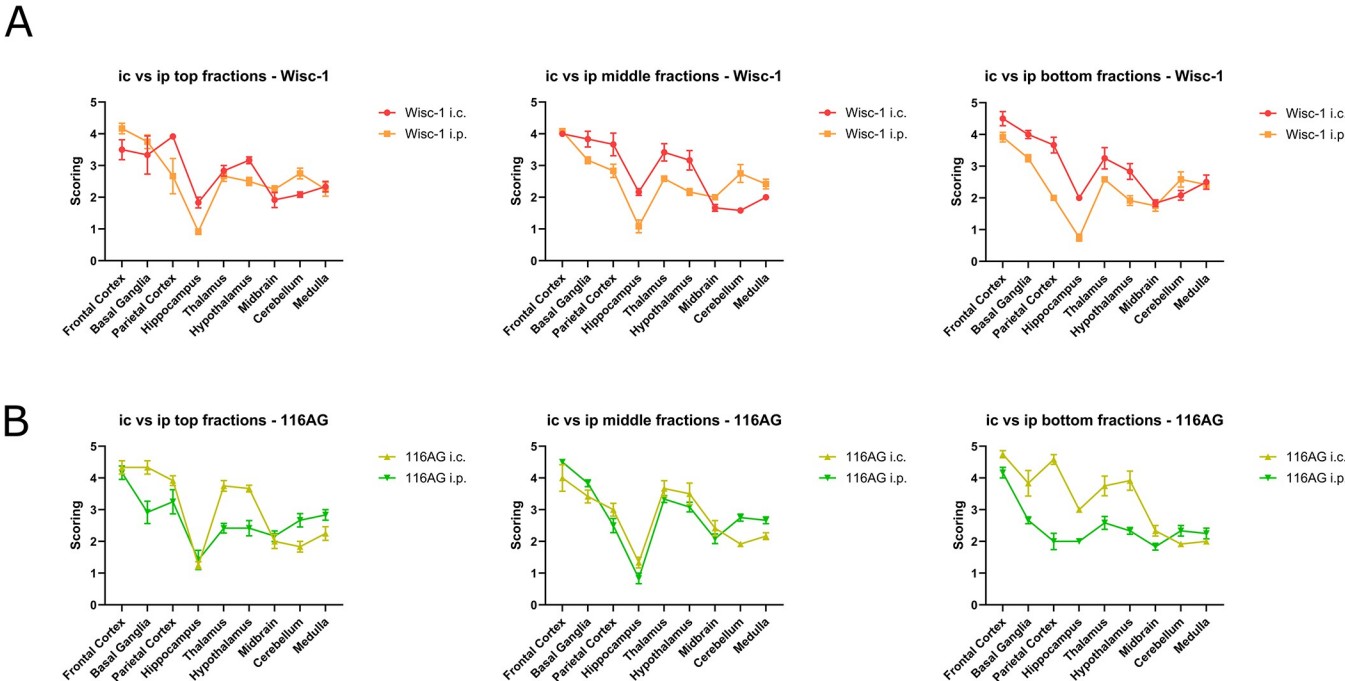

**Fig 5. Immunohistochemical analyses of *Prnp*.Cer.Wt brain homogenates inoculated i.c. or i.p. with Wisc-1 or 116AG fractions.** (A) PrP^Sc deposition scoring comparison of mice inoculated i.c. vs. i.p. with Wisc-1 top, middle, or bottom fractions. (B) PrP^Sc deposition scoring comparison of mice inoculated i.c. vs. i.p. with 116AG top, middle, or bottom fractions. Brains of n = 2 mice/group were analyzed. Error bars: SEM.

selection upon peripheral infection. These studies are important to understand CWD strain diversity and emergence in natural settings, which is critical to assess the risk of cross-species transmission of CWD.

Inoculation of a deer PrP^C overexpressing transgenic mouse model with Wisc-1 and 116AG CWD prions resulted in distinct survival times as well as PK resistance and conformational stability of PrP^Sc [28,29]. In our new gene-targeted mouse models, inoculated i.c. with Wisc-1 and 116AG prions we observed differences in PrP^Sc conformational stability, higher PrP^Sc quantity and deposition scores, and more pronounced spongiform lesions, compared to i.p. infection. First, the observed biochemical differences in i.c. inoculated mice confirm the presence of different strains in the Wisc-1 and 116AG prion isolates [26,28,29]. Second, differences in PrP^Sc quantity and distribution upon i.p. or i.c. infection likely reflect different pathways of prion spread within the brain. Upon i.c. inoculation, prions distribute via the interstitial system, facilitating seeding in different brain areas along with strain-specific selection of areas permissive for prion conversion and propagation. Upon i.p. infection, prions invade the brain predominantly from the spinal cord through caudal brain areas [48]. This could explain the similarly pronounced PrP^Sc deposition in medulla and cerebellum of i.c. and i.p. inoculated mice. From those areas, prions spread in a caudal-to-rostral direction through connective brain regions via the central autonomic network. We observed the greatest discrepancy in PrP^Sc deposition between i.c. and i.p. inoculated animals in cerebral areas, in line with previously described lower levels of deposition in the rostral brain areas at the time of terminal disease [48].

It is notable that upon i.p. infection, prions with identical conformational stability, and similar distribution and vacuolation profiles emerged from deer CWD isolates that, when inoculated i.c. into deer PrP^C overexpressing mice propagated as distinct and stable strains and substrains, Wisc-1/CWD1, and 116AG (short and long), respectively [25,26,28,29], with no evidence of common strain components [29]. Thus, we conclude that peripheral infection

represents a pre-selection step that can result in detectable propagation of minor substrain components present even in cloned strains [49]. Our findings also indicate that upon i.c. inoculation, substrains in a given isolate might compete and therefore, result in a more variable outcome. The controlled spread upon i.p. infection [48] also can restrict the variability of PrP<sup>Sc</sup> conformations propagated in the given environment and explain the generation of PrP$^{Sc}$ with identical conformational stability in brains of mice infected i.p. with either inoculum. Our results also indicate that distinct CWD strain mixtures found in the brains of cervids carrying PrP gene polymorphisms, such as the 116AG isolate, can evolve *in situ* in the brain due to differences in PrP primary structure. These conformations might not be stable upon natural, peripheral transmission, or propagate, in some cases, as a minor component in the brain. The replication in lymphatic tissues is more promiscuous [50], preserving conformational diversity of prions upon peripheral infection.

Our analysis of conformational stability of spinal cord tissues aimed at elucidating the bottleneck for conformational selection in the process of neuroinvasion. In particular, the finding that Wisc-1 and 116AG prions upon both i.p. and i.c. inoculation resulted in different PrP$^{Sc}$ conformational stability in this tissue shows that both isolates can propagate with greater diversity upon i.p. infection, reflecting the distinct PrP$^{Sc}$ conformations present in the inocula. More striking, however, are the comparisons between the two CNS tissues for the different routes of infection. In i.c. inoculated animals, both Wisc-1 and 116AG animals possessed prions in brain and spinal cord with virtually indistinguishable conformational stabilities. In contrast, i.p. inoculated animals generated prions with higher conformational stabilities in the spinal cord than the brain, independent of the CWD isolate used for infection. Notably, we collected entire spinal cords at terminal stage of disease when prions in the spinal cord are likely composed of both anterogradely and retrogradely transported PrP$^{Sc}$. Therefore, the conformational stability of prions that are transported to the brain in i.p. inoculated mice might be even higher than the values determined here, which encompass prions from both transport directions. CWD prion isolates from deer lymphoid tissue samples have been shown to possess greater variance in conformational stability than those from deer obex samples [51]. The authors in this previous study suggested a model where lymphogenic prions exist with greater strain diversity due to extraneural prion receptors permitting the propagation of various prion strains, and that prions transported to the brain undergo a more selective propagation process, with only a subset of conformations being able to replicate within the CNS, implying that selection occurs at the lymphatic tissue-CNS interface. However, the conformational stability properties of spinal cord and brain prions from i.p. inoculated animals indicate that conformational selection also takes place at the spinal cord-brain interface, or within the brain. First, we observe a reduction in prion conformational stability in brain compared to spinal cord, which is in line with a previous study describing highly neuroinvasive prions as conformationally unstable [38]. Furthermore, conformational differences between brain and spinal cord prions appear to depend on route of inoculation and with that, on direction of axonal transport, i.e., retrograde upon i.p. infection and anterograde upon i.c. infection. In i.p. infection, assuming that PrP$^{Sc}$ in spinal cord is predominantly from retrograde transport, prions undergo dynamic conformational adaptation or selection between spinal cord and brain. In i.c. infected mice, assuming that spinal cord prions originate mostly from anterograde transport, the virtually identical denaturation profiles of brain and spinal cord prions suggest that prion conformational properties do not change. These novel insights are important to consider in understanding the mechanism of prion neuroinvasion, as well as to the paradigm that prions of peripheral origin are less restricted in their ability to colonize lymphoreticular tissues [50,52], as our results suggest that additional conformational selection takes place beyond the lymphatic system.

Small oligomeric and soluble forms of PrP$^{Sc}$ were shown to harbor the most specific infectivity upon intracerebral inoculations in various models, with discrete strain-specific differences [41–44]. Here, we found that, in fact, the survival times of mice inoculated i.p. with a range of PrP$^{Sc}$ quaternary structures are significantly less variable than those of mice inoculated i.c. with the same PrP$^{Sc}$ aggregates. This finding is consistent for both CWD strains. It is important to note that here, we restrict our interpretation to comparing profiles of survival times across fractions within each inoculum. Variations with respect to relative amount of PrP$^{Sc}$ in each fraction are identical in i.c. and i.p, inoculation. Therefore, we argue that for comparing profiles and variance within each inoculum it is not necessary to characterize infectivity or seeding activity titers of individual fractions. IHC analysis indicates difference in PrP$^{Sc}$ distribution between fractions of i.c. inoculated mice as described earlier [44]. Variations in PrP$^{Sc}$ deposition patterns observed for the different route of inoculation for identical fractions mirror the findings in mice infected with whole brain homogenates. The overall greater variability in mice inoculated with fractionated 116AG prions, in particular in the bottom fraction, might indicate the presence of a substrain component [29]. Our results suggest that PrP$^{Sc}$ aggregation state is not a relevant factor governing infectivity of prions upon i.p. infection. This is in contrast to previous studies describing soluble, nonfibrillar aggregates as more neuroinvasive [39,53]. However, in these studies different routes of inoculation were used, with efficiency of prion uptake into neurons being the predominant measure upon inoculation into the tongue. Prion uptake by neurons is more efficient for smaller aggregates, shown also in cultured neuronal cells [54]. Upon i.c. infection, oligomeric PrP$^{Sc}$ aggregates harbor the highest level of infectivity [41–44]; however, in contrast to i.p. inoculation, this infection route bypasses the initial replication step in the spleen and interactions with immune cells. Spleen tissue expresses about 20-fold lower levels of PrP$^{C}$ than the brain [50]. This possibly results in enhanced interaction of PrP$^{C}$ with a wide range of PrP$^{Sc}$ aggregates due to reduced steric hindrance that might occur if PrP$^{C}$ molecules are clustered on the cell surface of cells with higher expression levels, attenuating variability between PrP$^{Sc}$ aggregate replication efficiency. Degradation of PrP$^{Sc}$ aggregates by immune cells [34,55,56] might decrease highly infectious oligomeric PrP$^{Sc}$ aggregates but enhance infectivity of fibrillary PrP$^{Sc}$ by fragmentation, and further mitigate differences observed upon i.c. inoculation.

Potential limitations of our study are that we analysed only two CWD isolates because of their well-characterized strain properties, and confirmation of our findings with additional CWD isolates would strengthen our conclusions. Furthermore, additional serial i.c. passages of i.p. passaged material would have been interesting to determine whether strain properties are maintained, or revert back to the original strain phenotype observed upon i.c. inoculation, but were beyond the scope of the current study.

In summary, our novel findings have important implications for the perception of CWD strain stability and transmission in cervid populations where peripheral infection is the predominant route of infection. We conclude that PrP$^{Sc}$ conformational selection occurs at the transition from spinal cord to brain in peripheral infection, but conformational diversity can evolve in the brain, driven e.g., by *Prnp* polymorphisms and other species- or tissue-specific factors.

## Materials and methods

### Ethics statement

This study followed the guidelines of the Canadian Council for Animal Care, and we performed all animal experiments in the study in compliance with the University of Calgary Animal Care Committee under protocol numbers AC18-0047 and AC22-0015.

### Prion isolates

Wisc-1 [26,27] and 116AG [28,29] CWD isolates originating from white-tailed deer expressing the wild-type deer PrP (Wisc-1) and from an animal heterozygous at codon 116 (116AG), respectively, were previously characterized. For fractionation by sedimentation velocity centrifugation, samples were generated by passaging the respective CWD strains to the 4th passage in tg(CerPrP132M)1536$^{+/+}$ mice overexpressing wild-type deer PrP [29]. Samples were prepared at a final concentration of 20% (wt./vol.) in phosphate-buffered saline (PBS; Life Technologies, Gibco) from deer obex samples or whole mouse brains and spinal cords using the MP Biomedicals fast prep-24 homogenizer (Fisher).

### Animal bioassay

Gene-edited mice possessing wild-type deer PrP (*Prnp*.Cer.Wt) were described, and brain and spinal cord homogenates of mice inoculated i.c. and i.p. with unfractionated Wisc-1 and 116AG prions from a previous study [5] were analysed here. For analysis of fractionated CWD, female mice between six to eight weeks old were anaesthetized and i.c. inoculated with 20 µl of the CWD fractions obtained upon sedimentation velocity centrifugation in the right parietal lobe using a stainless-steel injection needle 0.60 mm diameter x 4 mm (Unimed S.A.). I.p. inoculation was done with 100 µl of the CWD fractions with a 27-gauge needle (BD). (n = 5 mice per fraction and route of inoculation). The mice were monitored weekly until the onset of clinical signs (examples of which include ataxia, imbalance, paralysis, kyphosis, and weight loss), and then monitored daily until they reached terminal prion disease. The animals were anesthetized before being euthanized by isofluorane overdose followed by cardiac exsanguination. Tissue samples were collected and frozen at -80˚C and/or fixed in 10% formalin for further use.

### Sedimentation velocity centrifugation

Sedimentation velocity ultracentrifugation of the prion samples was done as described previously [42–44]. Briefly, 100 µl of 10% brain homogenate was solubilized in an equal volume of solubilization buffer (50 mM HEPES pH 7.4, 300 mM NaCl, 10 mM EDTA, 2 mM DTT, 4% (wt/vol.) dodecyl-β-D-maltoside; Sigma) on ice for 45 minutes. Sarkosyl (N-lauryl sarcosine; Sigma) was added to a final concentration of 2% (wt/vol.) and the incubation continued for another 45 minutes on ice. The sample was loaded onto a continuous 10–25% iodixanol gradient (Optiprep; Sigma), the linearity of which was verified by refractometry. The gradients were centrifuged at 285,000 *g* for 45 minutes at 4˚C in a SW-55 rotor using an Optima XE-90 ultracentrifuge (Beckman Coulter). Thirty fractions of 160 µl were collected from the top to the bottom for Western blot characterization or mouse bioassays.

### Proteinase-K (PK) digestion

Aliquots of 10% brain homogenate were digested with either a range of PK concentrations from 50 µg/ml to 5000 µg/ml, or with a fixed concentration of 50 µg/ml of PK in the presence of lysis buffer at 37˚C for one hour. Pefabloc protease inhibitor (1x; VWR) was added to terminate the enzymatic reaction, followed by the addition of 3x SDS sample loading buffer.

### SDS-PAGE and Western blot

Samples were separated on a 12.5% SDS-poly-acrylamide gel and transferred to PVDF membranes (Amersham, GE Healthcare). Membranes were blocked with 5% non-fat milk in Tris-buffered saline with a final concentration of 0.1% Tween-20 (TBST) before being probed with the anti-PrP monoclonal antibody 4H11 (1:1000) [57] followed by horseradish peroxidase-

conjugated goat anti-mouse IgG antibody (Sigma) and developed using Luminata horseradish peroxidase substrate (MilliporeSigma). Images were acquired using the ChemiDoc imaging system (Bio-Rad)), and the Image Lab (Bio-Rad) software was used to quantify the PrP signals on the respective digital images. Calculations were done on Microsoft Excel, and graphs generated using GraphPad Prism 10.

## Conformational stability assay

Brain homogenate samples were incubated with guanidine hydrochloride (GdnHCl; 0–4 M) for 1 hour at room temperature. Each sample was then adjusted to 0.5 M GdnHCl, followed by digestion with 5 μg/ml of PK in the presence of cell lysis buffer (10 mM Tris-HCl pH 7.5, EDTA, 100 mM NaCl, 10 mM EDTA, 0.5% sodium deoxycholate, 0.5% Triton X-100) for one hour at 37°C, and 1x Pefabloc protease inhibitor (VWR) was added to terminate the enzymatic reaction. The samples were then transferred onto nitrocellulose membrane (Pall) using a dot blot apparatus (Bio-Rad). Following with 5% non-fat milk in TBST, the membrane was incubated with anti-PrP monoclonal antibody 4H11 (1:1000) [57] followed by horseradish peroxidase-conjugated goat anti-mouse IgG antibody (Sigma) and developed using Luminata horseradish peroxidase substrate (MilliporeSigma). Images were acquired using the ChemiDoc imaging system (Bio-Rad), and the Image Lab (Bio-Rad) software was used to quantify the PrP signals on the respective digital images. Figure displays were generated using GraphPad Prism 10, with the half-maximal GdnHCl value (GdnHCl$_{50}$) calculated using non-linear regression, using a sigmoidal dose-response inhibition equation with a variable slope (four parameters) setting. Tissue samples from n = 6 (i.c.) and n = 8 (i.p.) animals per group were each assayed between one and three times for a minimum of nine replicates per group.

## Immunohistochemistry

Immunohistochemical analysis was performed with reference to previous publication with modifications [58]. Serial sections of formalin-fixed and paraffin-embedded brains were cut at 4 μm thickness and 2–4 consecutive sections were used for analysis. They were autoclaved (2.1 × 10$^5$ Pa) for 25 minutes in citrate buffer (10X Target Retrieval Solution, pH 6.0; Agilent Dako, Dako S1699) at 121°C, followed by incubation in 98% formic acid (Sigma) for 5 minutes. Sections were further blocked in 5% normal goat serum (Dako X0907) for 15 minutes and incubated overnight at 4°C with a primary anti-PrP monoclonal antibody Bar224 (epitope aa 144–154, cerPrP numbering; 1:2000; Cayman Chemical). A highly sensitive detection system using HRP-linked secondary antibody (EnVision+ HRP Labelled Polymer Anti-Mouse, Dako K4001) was used, sections were developed using 3, 3'-diaminobenzidine tetrahydrochloride (DAB) and counterstained with Gill II hematoxylin. Slides were scanned using the NanoZoomer 2.0-RS digital slide scanner (Hamamatsu Photonics K.K.) and images were visualized and read in the NDP.view 2 software (Hamamatsu). PrP$^{Sc}$ distribution was scored at nine different regions of the brain (frontal cortex, basal ganglia, parietal cortex, hippocampus, thalamus, hypothalamus, midbrain, cerebellum, and medulla/pons) on a scale of 0 (absence of PrP$^{Sc}$ staining) to 5 (widespread PrP$^{Sc}$ staining). Scoring was performed blindly over three separate times per analyzed brain sample of n = 2 mice per individual group. The scores are reported as the mean ± standard error of means of repeated readings for individual animals and the values were plotted by using GraphPad Prism 10.

## Vacuolation profile

Vacuolation scoring was performed as described previously [59, 60]. Sagittal brain sections (5 μm-thick) were cut and 2 consecutive sections were stained using haematoxylin and eosin

and analysed. Slides were scanned using the NanoZoomer 2.0-RS digital slide scanner (Hamamatsu Photonics K.K.) and images were analyzed using the NDP.view 2 software (Hamamatsu). The extent of spongiform degeneration was assessed in 9 brain regions (frontal cortex, basal ganglia, parietal cortex, hippocampus, thalamus, hypothalamus, midbrain, cerebellum, and medulla/pons). Presence and severity of spongiform degeneration was scored on a scale from 0–5, with a score of 0 describing the absence, and a score of 5 the most severe lesions [60]. Scoring was performed in brains of n = 2 mice per individual group at least 6 times in a blinded manner. The scores are reported as the mean ± standard error of means of repeated readings for individual animals and the values were plotted by using GraphPad Prism 10.

## Statistical analysis

Mean with the standard deviation were reported for bioassay survival figures, while mean with standard error of means were reported for all other parameters. Statistical analyses were performed with GraphPad Prism 10. Comparisons involving two groups were done using unpaired Student's *t*-test unless otherwise stated. Comparisons involving three or more groups were done with one-way ANOVA followed by post hoc analysis with Tukey's multiple comparison test.

## Supporting information

**S1 Fig. Proteolytic digestion analyses of *Prnp*.Cer.Wt brain homogenates inoculated i.c. or i.p. with Wisc-1 or 116AG prions.** (A) Representative blots of PrP$^{Sc}$ for individual animals from each group digested with various concentrations of PK; n $\geq$ 9 over a minimum of 5 animals per group. (B) Densitometric analysis of the PrP$^{res}$ signals. Signals were quantified as a ratio of the baseline signal at 50 μg/ml. Statistical analysis was performed with two-way ANOVA followed by Tukey's post-hoc multiple comparison test. * *p*-value = 0.0369. (C) cPK$_{50}$ analysis of the PrP$^{res}$ signals of the three groups. The PK concentration required to degrade 50% of the signal was obtained from the baseline signal of 50 μg/ml digestion. Statistical analysis was performed with one-way ANOVA followed by Tukey's multiple comparison test. ** *p*-value = 0.0042. n $\geq$ 13, error bars: SEM.
(TIFF)

**S2 Fig. Immunohistochemical and vacuolation analyses of *Prnp*.Cer.Wt mouse brain tissue inoculated i.c. or i.p. with Wisc-1 or 116AG prions.** (A) PrP$^{Sc}$ deposition scoring comparison of Wisc-1- vs. 116AG- inoculated mice through different routes. (B) PrP$^{Sc}$ deposition scoring comparison of Wisc-1 i.c. vs. i.p. inoculated groups. (C) PrP$^{Sc}$ deposition scoring comparison of 116AG i.c. vs. i.p. inoculated groups. (D) brain lesion profile comparison of Wisc-1 vs. 116AG inoculated mice through different routes. (E) Brain lesion scoring comparison of Wisc-1 i.c. vs. i.p. inoculated groups. (F) Brain lesion scoring comparison of 116AG i.c. vs. i.p. inoculated groups. Brains of n = 2 mice/group were analysed. Error bars: SEM.
(TIFF)

**S3 Fig. Representative histological slices depicting PrP$^{Sc}$ deposits at various brain regions for each group of mice.** Brain samples from 2 mice were analyzed for each group, and scoring was performed blindly three times for each sample.
(TIF)

**S4 Fig. Representative dot blot of PrP$^{Sc}$ from brain and spinal cord tissues denatured with various concentrations of GdnHCl followed by PK digestion.**
(TIFF)

**S5 Fig. Proteolytic digestion of *Prnp*.Cer.Wt brain homogenate inoculated i.c. or i.p. with Wisc-1 or 116AG fractions.** Representative blots of PrP$^{Sc}$ from mice inoculated i.c. or i.p. with (A) Wisc-1 or (B) 116AG fractions digested with 50 μg/ml of PK.
(TIFF)

**S1 Data. Extended supporting information.**
(XLSX)

## Acknowledgments

Portions of the manuscript were developed from the theses of MIA and SCC. We thank Yuanmu Fang and Kristina Santiago-Mateo, Canadian and WOAH reference lab for BSE, CFIA, Lethbridge, Canada, for performing IHC experiments, Dr. Debbie McKenzie (University of Alberta) and Dr. Trent Bollinger (Canadian Wildlife Health Cooperative) for providing the Wisc-1 and 116AG CWD isolates, respectively. We are grateful to staff from the University of Calgary Clara Cristie Mouse Genomics Center and the Prion-Virology Animal Facility for outstanding animal care.

## Author Contributions

**Conceptualization:** Sheng Chun Chang, Sabine Gilch.

**Data curation:** Sheng Chun Chang.

**Formal analysis:** Sheng Chun Chang, Maria Immaculata Arifin, Hermann M. Schatzl, Sabine Gilch.

**Funding acquisition:** Hermann M. Schatzl, Sabine Gilch.

**Investigation:** Sheng Chun Chang, Maria Immaculata Arifin, Waqas Tahir, Keegan John McDonald, Doris Zeng, Samia Hannaoui.

**Methodology:** Waqas Tahir.

**Project administration:** Sabine Gilch.

**Resources:** Sabine Gilch.

**Supervision:** Samia Hannaoui, Sabine Gilch.

**Validation:** Sheng Chun Chang, Maria Immaculata Arifin, Sabine Gilch.

**Visualization:** Sheng Chun Chang, Maria Immaculata Arifin.

**Writing – original draft:** Sheng Chun Chang, Sabine Gilch.

**Writing – review & editing:** Sheng Chun Chang, Maria Immaculata Arifin, Waqas Tahir, Keegan John McDonald, Doris Zeng, Hermann M. Schatzl, Samia Hannaoui, Sabine Gilch.

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
