## [Decision Letter · Decision Letter 0]

25 Apr 2024

Dear Dr. Gilch,

Thank you very much for submitting your manuscript "Extraneural infection route restricts prion conformational variability and attenuates the impact of quaternary structure on infectivity" for consideration at PLOS Pathogens. As with all papers reviewed by the journal, your manuscript was reviewed by members of the editorial board and by several independent reviewers. The reviewers appreciated the attention to an important topic. Based on the reviews, we are likely to accept this manuscript for publication, providing that you modify the manuscript according to the review recommendations.

Sincerely,

Amanda L. Woerman

Academic Editor

PLOS Pathogens

Neil Mabbott

Section Editor

PLOS Pathogens

Michael Malim

Editor-in-Chief

PLOS Pathogens

orcid.org/0000-0002-7699-2064

Reviewer Comments (if any, and for reference):

Reviewer's Responses to Questions

**Part I - Summary**

Reviewer #1: The parameters that control prion strain diversity are poorly understood, especially for animals infected with chronic wasting disease (CWD) prions. This study utilized gene targeted mice expressing cervid PrP to investigate the route of prion infection on prion strain diversity. The authors found that two distinct prion strains, Wisc -1 and 116AG that maintained differences in strain properties following i.c. inoculation. Interestingly, intraperitoneal (i.p.) inoculation resulted in a change in strain properties that included incubation time, neuropathology and, most importantly, PrPSc conformational stability. The authors show differences in PrPSc conformational stability in spinal cord and brain homogenates in the same animal. Finally, using sedimentation gradient fractionation studies, the authors provide evidence that quaternary PrPSc structure does not contribute to strain variation. Overall, this is an important observation that has significant implications on CWD strain evolution in the natural host and the design of laboratory experiments of prion-infected rodents. There are, however, several areas where additional analysis and clarification would aid the reader and strengthen the conclusions of the authors.

Reviewer #2: The paper describes the transmission of two well-characterised and distinct strains of CWD (Wisc1 and 116AG) to a gene-targeted mouse line expressing wt (consensus) deer PrP at physiological levels. Interestingly, the authors demonstrate that the route of infection (intraperitoneal versus intracerebral injection) has an effect on the expression of certain strain-specific characteristics (lesion profile, conformational stability), which provide evidence that peripheral infection (i.p.) may result in selection of convergent sub-strains capable of neuroinvasion. Fractionation of PrPSc from the two strains based on aggregate size, had less impact on transmission and strain characteristics when injected by i.p. compared with i.c. route.

This study is important, since the majority of prion strain characterization experiments have been carried out in over-expressing transgenic mouse lines, rather than gene-targeted models. There is increasing recognition that over-expressing lines may not accurately recapitulate all features of natural disease pathogenesis, particularly peripheral pathogenesis involving lymphoid tissue and the peripheral nervous system. Therefore, this study has performed an in-depth comparison of infection by the i.c. and i.p. (peripheral routes) and made several interesting and novel observations. Most intriguingly, the authors demonstrate that spinal cord PrPSc has greater conformational stability than brain PrPSc in i.p. infected mice only, suggesting that conformational selection may be occurring within the central nervous system, as well as in peripheral tissues.

The paper is well-written and the work has been carefully performed and is generally clearly described and illustrated in the figures and supplementary information.

Reviewer #3: The authors demonstrated that the two distinct chronic wasting disease strains were similar when injected intracerebrally but differ within in the brain and spinal cord when injected intraperitoneally further proving CWD extra neuronal propagation is unique. These findings are very interesting and show that the more natural mode of transmission, peripheral exposure, is highly stable conformationally within the brain. Many of these findings are based on immunohistochemistry and morphological changes however this analysis was performed on a small sample size and would need to be increased in order to make these claims.

**Part II – Major Issues: Key Experiments Required for Acceptance**

Reviewer #1: (No Response)

Reviewer #2: (No Response)

Reviewer #3: 1.) Figures 2 and 5 are showing pathological scoring data, for both PrPSc deposits and vacuolation, of different brain regions in brain samples from a minimum of 2 mice for each group. The number of brains analyzed should increase for rigor of the analysis to at least an N of 3 or 4 to properly interpret this data. Currently this is too small of a sample size to make any conclusions.

2.) The fractionated PrPSc was shown to decrease variance in survival with ip indicating that peripheral infection is independent of quaternary structure. This is a significant and interesting finding. Would other peripheral organs (ie. spleen or lymph) have a similar PrPSc immunoblot and conformational stability profile as the brain and spinal cord in the ip inoculated animals? This would help to explain if this is a nervous tissue or periphery vs central finding. This data would be beneficial to understand more about the tissue specific factors involved.

**Part III – Minor Issues: Editorial and Data Presentation Modifications**

Reviewer #1: In general, the authors are encouraged to provide p values where significance is indicated and to provide information on the number of animals examined for each experimental group.

The observed differences in incubation time, PrPSc abundance and neuropathology between i.c. and i.p. infected animals could be due to route of infection and not necessarily a change in strain properties. However, the PrPSc conformational stability data argues against this explanation and is very intriguing. In the manuscript the authors present this data in figure 3. It is unclear from the material and methods or the results section how many animals per each group were tested or how many technical replicates were performed. Additionally, the GdnHCl half values and SEM are not reported or are statistical significance p values reported. Inclusion of these would strengthen the conclusions of the authors. Additionally, in figure 3, panels E and F, the 0M values are lower in spinal cord vs. brain. Do the authors think differences in the starting amount of PrPSc is contributing to the differences observed?

Regarding the spinal cord, additional details on the collection of the spinal cord are important for a complete understanding of lines 276-286. What level of the spinal cord was collected and did this vary between animals? In the discussion, the authors suggest that strain adaptation is occurring in the spinal cord prior to transport to the brain. Based on the anatomy of neuroinvasion following i.p. inoculation this would be expected to occur in the IML of the thoracic spinal cord, replication in these cells followed by retrograde axonal transport to the brain. Clarification of the spinal cord collection will mitigate this concern. Also, at terminal disease the cord from i.p. infected animals will contain a mixture of IML originating prions and those that migrate from the brain, the authors should consider this in the interpretation as the effect in the IML may be greater than observed in the cord homogenate.

Two additional pieces of data, if available, would further strengthen the manuscript. The idea of prion adaptation during the pathogenesis of disease is very interesting and important, do the authors have any insight into what is occurring in the spleen? Second, have the authors taken the i.p. passaged material and performed a serial i.c. passage to see if the strain properties are maintained or if it reverts back to the original strain phenotype?

Reviewer #2: 1. It would be helpful if the source of unfractionated inoculum for the initial transmission experiments could be stated – in particular, the genotype of the WTD infected with either Wisc1 or 116AG.

2. Although Wisc1 and 116AG are described as being distinct strains of CWD, their incubation periods and distribution of PrPSc/vacuolation in the brain are quite similar in Prnp.Cer.Wt mice, whether injected i.c. or i.p. The major strain-specific differences between mice inoculated i.c. seem to be in the vacuolation profile (Fig 2D), and in PrPSc conformational stability/relative PK resistance. Can the authors comment on whether these differences reflect the strain-specific characteristics observed when these specific CWD strains are passaged in mouse lines over-expressing CerPrP?

3. The scoring system for PrPSc deposition in different brain regions is not described in detail in the methods, nor citation provided for a published method. The methods state that the images were analysed using NDP.view 2 software – how? By eye? If so, how are the different scores specifically defined (does it take account of different types of staining e.g. plaques, intraneuronal, glial, ependymal etc), and is a known reference slide included in each IHC run as a comparator to account for inter-run variability in staining? Each analysed sample was scored blindly three times – was all scoring done by one individual, or was more than one person responsible (similar questions for vacuolation scoring)?

4. Figure 2 – it would be helpful if the legend could state the numbers of animals analysed to provide the scores. Was any statistical analysis performed to determine the significance of observed differences? Error bars are stated as SEM, but in the methods it says that scores are reported as mean ± standard deviation – please clarify.

5. Conformational stability of PrPSc (Figure 3) - Values are expressed as a ratio of baseline signal at 0M Gdn-HCl – this suggests that all curves should start at a value of 1, and the fact that some do not makes it more difficult to compare curves in some cases. Can the authors explain why this is? Also, although an indication of statistical methods and key to p values is provided in the legend to Fig 3, there are no symbols (asterisks) within the figure charts to indicate where statistically significant differences were found.

6. The results with the fractionated brain homogenates are somewhat puzzling. The most likely explanation for the variation in incubation periods observed with ic inoculation of fractions would be differences in titre, and to some extent this seems to fit expectation, with the shortest IPs for fractions with relatively small aggregates, and longer IPs for fractions containing larger aggregates. If this is the case, a similar pattern might be expected in the mice inoculated ip with the same fractions, but this is not the case. Could this be a dose effect? In that ip inoculated mice receive a much larger dose (volume)? Or does it suggest that favourable conditions for peripheral selection of a neuroinvasive sub-strain of ip injected prions exist across a wide range of infectivity/PrPSc titres? Were end-point titrations of Wisc1 and 116AG by ic and ip injection performed in Prnp.Cer.Wt mice at any point, and if so, could this be helpful in relating incubation periods to titre for the fractionated brains?

7. Figure 5 – IHC profiles from mice inoculated with different brain fractions. The authors point out that for Wisc1, the profiles are fairly consistent across fractions in mice inoculated by both routes, whereas there is greater variability for 116AG, particularly for the bottom fraction. Can they comment on whether they consider this to be due to isolation of different strains or sub-strains in the different fractions of 116AG? Do lesion profiles from scoring vacuolation give similar results? Again, would be helpful to have the numbers of mice represented in the plots indicated in the figure legend (n=?).

8. In the discussion (lines 306-309), the authors suggest: “Spleen tissue expresses about 20-fold lower levels of PrPC than the brain (50). This possibly results in better accessibility and enhanced interaction of PrPC with a wide range of PrPSc aggregates, attenuating variability between PrPSc aggregate replication efficiency.” It is hard to follow the reasoning – would a lower concentration of PrPC not suggest a reduced number of sites for PrPSc/PrPC interactions? This could be clarified. Is it possible that competition between (sub)-strains within the brains of ic injected mice, leads to greater variability in outcomes, whereas the neuroinvasive sub-strain selected during ip infection can replicate more consistently and reproducibly when it reaches the brain?

Reviewer #3: 1.) Is the deposition of PrPSc found equally on both hemispheres of the brain or is one hemisphere more affected than the other? Does this differ between ip and ic inoculations. This information can also add information about these two different isolates and the possible different strains ic and ip are producing.

2.) It would be beneficial to discuss the implications and possible mechanistic/ pathological reasons the Wisc-1 isolate survival time did not correlated to amount of PrPSc. This is an important difference found in these two relatively similar isolates and should be discussed further.

3.) It would be important to test this extaneuronal phenom

---

## [Decision Letter · Decision Letter 1]

10 Jun 2024

Dear Dr. Gilch,

Thank you very much for submitting your manuscript "Extraneural infection route restricts prion conformational variability and attenuates the impact of quaternary structure on infectivity" for consideration at PLOS Pathogens. As with all papers reviewed by the journal, your manuscript was reviewed by members of the editorial board and by several independent reviewers. The reviewers appreciated the attention to an important topic. Based on the reviews, we are likely to accept this manuscript for publication, providing that you modify the manuscript according to the review recommendations.

The concerns initially raised by Reviewer 3 regarding sample size for neuropathology have not be adequately addressed, as Reviewer 1 now indicates. The concerns raised by Reviewer 1 are important to address prior to manuscript acceptance.

Sincerely,

Amanda L. Woerman

Academic Editor

PLOS Pathogens

Neil Mabbott

Section Editor

PLOS Pathogens

Michael Malim

Editor-in-Chief

PLOS Pathogens

orcid.org/0000-0002-7699-2064

The concerns initially raised by Reviewer 3 regarding sample size for neuropathology have not be adequately addressed, as Reviewer 1 now indicates. The concerns raised by Reviewer 1 are important to address prior to manuscript acceptance.

Reviewer Comments (if any, and for reference):

Reviewer's Responses to Questions

**Part I - Summary**

Reviewer #1: The authors have addressed most of the issues raised by the reviewers. One major issue, however, remains and needs to be addressed

Reviewer #2: See previous comments

Reviewer #3: The authors thoroughly justified the major and minor issues of the reviewer. I believe the manuscript should be accepted with the completed revisions.

**Part II – Major Issues: Key Experiments Required for Acceptance**

Reviewer #1: The responses from the authors to the concerns regarding the data in figure 2 raise concerns regarding rigor. For the vaculation profile a n of two animals is insufficient to make meaningful conclusions. It is unclear how a SEM can be represented in the figure from an n of two. The differences measured could simply reflect animal variation and conclusions based on this are suspect.

Second, for the PrPSc deposit data, the authors indicate in the rebuttal that "Between 2 and 4 sections were read 3 times from each mouse brain and averaged." The number of reads is fine, but examining 2-4 sections per animal is inadequate. Subtle differences in the anatomical location between the sections and differences in IHC performance could explain differences rather than a true difference in PrPSc distribution. Overall, conclusions based on figure 2 in its current form are not completely supported.

Reviewer #2: See previous comments

Reviewer #3: (No Response)

**Part III – Minor Issues: Editorial and Data Presentation Modifications**

Reviewer #1: (No Response)

Reviewer #2: See previous comments

Reviewer #3: (No Response)

PLOS authors have the option to publish the peer review history of their article (what does this mean?). If published, this will include your full peer review and any attached files.

Reviewer #1: No

Reviewer #2: No

Reviewer #3: No

Figure Files:

Data Requirements:

Reproducibility:

References:

---

## [Editor Report · Decision Letter 2]

25 Jun 2024

Dear Dr. Gilch,

We are pleased to inform you that your manuscript 'Extraneural infection route restricts prion conformational variability and attenuates the impact of quaternary structure on infectivity' has been provisionally accepted for publication in PLOS Pathogens.

Best regards,

Amanda L. Woerman

Academic Editor

PLOS Pathogens

Neil Mabbott

Section Editor

PLOS Pathogens

Michael Malim

Editor-in-Chief

PLOS Pathogens

orcid.org/0000-0002-7699-2064
---

## [Editor Report · Acceptance letter]

2 Jul 2024

Dear Dr. Gilch,

We are delighted to inform you that your manuscript, "Extraneural infection route restricts prion conformational variability and attenuates the impact of quaternary structure on infectivity," has been formally accepted for publication in PLOS Pathogens.

Best regards,

Michael Malim

Editor-in-Chief

PLOS Pathogens

orcid.org/0000-0002-7699-2064